# Brugada Syndrome Associated with Different Heterozygous *SCN5A* Variants in Two Unrelated Families

**DOI:** 10.3390/jcm11195625

**Published:** 2022-09-24

**Authors:** Nadine Molitor, Argelia Medeiros-Domingo, Siv Fokstuen, Frank Ruschitzka, Firat Duru, Ardan Saguner

**Affiliations:** 1Department of Cardiology, University Heart Center Zurich, 8091 Zurich, Switzerland; 2Swiss DNAlysis, 8600 Dubendorf, Switzerland; 3Genetic Medicine Division, Diagnostic Department, Hôpitaux Universitaires de Genève, 1205 Geneva, Switzerland; 4Center for Integrative Human Physiology (ZIHP), University of Zurich, 8006 Zurich, Switzerland

**Keywords:** Brugada syndrome, channelopathy, *SCN5A*, arrhythmogenic right ventricular cardiomyopathy, overlapping syndrome

## Abstract

The cardiac sodium channel (Nav1.5) controls cardiac excitability by triggering the action potential of cardiac myocytes and controlling electric impulse transmission. However, it has also been associated with arrhythmogenic cardiomyopathies. Accordingly, genetic variants in *SCN5A* that result in loss of function of Nav1.5 are associated with inherited arrhythmia syndromes, which are caused by reduced cardiac excitability, particularly Brugada syndrome (BrS) as well as arrhythmogenic right ventricular cardiomyopathy (ARVC). We report a novel pathogenic *SCNA5* variant being associated with BrS overlapping with ARVC, as well as disease progression with a previously reported *SCN5A* variant being associated with a phenotype of BrS and conduction system disorder in two unrelated families.

## 1. Introduction

Brugada syndrome (BrS) is a hereditary cardiac disorder and has been linked to genetic variants, mainly in the *SCN5A* gene [1,2], located on the short arm of chromosome 3 encoding the α-subunit of the sodium channel NaV_1.5_ [3]. It is characterized by dynamic ST-segment elevations in leads V1-3, typically followed by negative T waves. An RBBB pattern of varying degrees is observed in some patients. The characteristic ECG pattern can be present spontaneously or unmasked by drugs [4,5]. The syndrome is associated with syncope and sudden cardiac death (SCD] caused by polymorphic ventricular tachycardia (VT) and ventricular fibrillation (VF) [2,6].

We report two unrelated families harboring different *SCN5A* variants associated with an autosomal-dominant form of BrS (one causative variant in each family). One variant is novel, while the other is a variant first described in 2005 [7], that reveals novel aspects in the same previously described family.

## 2. Materials and Methods

### 2.1. Genetic Analysis

DNA extraction was performed from an EDTA blood sample in both families using the Prepito (Perkin Elmer) DNA Blood250 kit. For molecular genetic analyses, the SwissDNAlysis Cardiopanel (Agilent, SureSelectQXTTarget Enrichment) was used for family 1 and high-throughput sequencing (Illumina MiSeq) was performed, with 95.1% of bases sequenced with a Q-score > Q30. 99.3% of the analyzed gene segments having a coverage of ≥20X. The average sequencing depth of the analyzed gene segments was 155.4X. Comprehensive genetic testing was limited to a panel of 16 BrS-associated genes (*CACNA1C*, *CACNA2D1*, *CACNB2*, *GPD1L*, *HCN4*, *KCND3*, *KCNH2*, *KCNE3*, *PKP2*, *RANGRF, SCN10A*, *SCN1B*, *SCN2B*, *SCN3B*, *SCN5A**, *TRPM4*), considering the evidence-based evaluation of gene validity for BrS by an expert panel in 2018 [3], and the latest 2022-ESC guidelines in genetic testing for inherited heart diseases (1), selected from a panel of 173 inherited heart disease genes the patient was screened for, and no other suspected variant able to produce the phenotype was detected in this case within those 173 genes.

In family 2, for molecular genetic analyses the Illumina TruSight Cardio was used with DNA prep with enrichment and high-throughput sequencing (Illumina MiSeq), with 91.8% of bases sequenced with a Q score ≥ Q30. Ninety-nine percent of the analyzed gene segments had coverage of ≥20X. The average sequencing depth of the analyzed gene segments was 358.9X. Comprehensive genetic testing was performed using a custom panel of 15 candidate genes, which at that time had been associated with BrS (*CACNA1C*, *CACNA2D1, CACNB2*, *GPD1L*, *HCN4*, *KCND3*, *KCNE3*, *PKP2*, *RANGRF*, *SCN10A*, *SCN1B*, *SCN2B*, *SCN3B*, *SCN5A**, *TRPM4*), selected from a panel of 173 inherited heart disease genes the patient was screened for, and no other suspected variant able to produce the phenotype was detected in this case within those 173 genes.

Alignment of the sequences and local realignment against the human reference genome (GRCh37hg19) were performed with lllumina alignment software v2.5.42.7 (Burrows–Wheeler algorithm and GATK for variant calling). Variants with an allele frequency <5% in the coding regions including the flanking intronic regions (±10 bp) were scored.

Variant Studio 3.0, Varsome Clinical, dbSNP153, gnomAD database, PubMed and ClinVar were used for data interpretation. All variants in this report were confirmed by conventional Sanger sequencing.

MLPA was used for relative copy number analysis to detect deletions and duplications (CNV analysis) in the *SCN5A* gene (up, exon 1 to 18, 19a and exon 20 to 29) (MRC Holland; SALSA P108 (B4-0819)). Data analysis was performed with Coffalyser.Net (v.210604.1451, MRC-Holland, Amsterdam, The Netherlands).

### 2.2. Clinical Evaluation at Our Center

In family 1, we obtained from the index patient a detailed medical and family history over 4 generations and performed a 12-lead ECG, flecainide challenge test, exercise stress testing, transthoracic echocardiography, 18F-FDG PET CT scan, programmed ventricular stimulation, and endocardial 3D electroanatomical voltage mapping of the right ventricle. His first-degree family members, who live abroad, are asymptomatic, but have not been referred for cardiologic workup and genetic cascade screening thus far. He has no children. In family 2, a detailed medical and family history was taken from the brother of the index patient. Moreover, a 12-lead ECG and flecainide or ajmaline challenge tests were performed in advance on all living first-degree family members, if no spontaneous BrS ECG pattern was visible on a spontaneous 12-lead ECG.

#### Electroanatomical Mapping in Family 1

Three-dimensional endocardial electroanatomical mapping was performed with the CARTO electroanatomic mapping system (Biosense Webster, Diamond Bar, CA, USA). A 7F Navi-Star Thermocool D-curve catheter (3.5 mm tip, Biosense Webster) was used as the mapping/ablation catheter. The catheter was carefully moved over the endocardial surface to record electrograms at different sites and to simultaneously determine the shape and volume of the right ventricle (RV) during sinus rhythm via a long deflectable sheath (Agilis, Abbott, Chicago, IL, USA). Unipolar and bipolar electrograms were recorded simultaneously.

## 3. Results

### 3.1. Family 1

A 37-year-old asymptomatic man (index patient) was diagnosed with a borderline BrS type I ECG during a routine cardiac checkup based on 12-lead ECG a few years ago (Figure 1). A flecainide challenge test, in the current absence of a spontaneous BrS ECG pattern, revealed drug-induced Brugada type 1 ECG changes. An invasive electrophysiological evaluation including three-dimensional endocardial electroanatomical mapping showed RV subtricuspid inferior low-voltage areas. Repetitive, sustained monomorphic VT could be induced from this region during isoprenaline provocation. A cardiac 18-FDG-PET-CT excluded cardiac sarcoidosis or myocarditis. The family history was negative for sudden cardiac death, unexplained syncope, drowning, near drowning, or single person motor vehicle accidents over four generations.

Comprehensive genetic testing revealed a novel pathogenic (according to the 2015 ACMG Criteria, Class V, PVS1, PM2, PP3) [8] heterozygous splice-site variant in the *SCN5A* gene (c.4814-1G > A; p. (?) in intron 27, which has not been described in the literature so far. A transvenous single-chamber ICD was implanted for primary prevention based on the Brugada ECG, a positive electrophysiological study with the pathogenic substrate, and the pathogenic *SCN5A* variant. The parents of the patient had not been tested so far.

### 3.2. Family 2

A 34-year-old man with a history of BrS diagnosed by 12-lead ECG underwent ICD implantation for primary prevention at the age of 15 years. His cardiac MRI was unremarkable. Since implantation, he experienced non-sustained ventricular tachycardia episodes not necessitating ICD therapies. His 37-year-old brother (index patient) was also diagnosed with BrS after surviving SCD caused by VF. He received an ICD for secondary prevention at the age of 16 years. The 39-year-old sister and their mother (67 years old) were asymptomatic. The father’s 12-lead ECG (67 years old) showed first-degree AV block (PQ 240 ms), left anterior fascicular block, and QRS prolongation (160 ms) (Figure 2), whereas the mother’s and sister’s 12-lead ECGs were unremarkable. An ajmalin challenge test in the parents of the index patient was unremarkable.

Comprehensive genetic testing was performed in the brother of the index patient at our center. The patient was found to be a heterozygous carrier of a pathogenic truncating non-sense variant in the *SCN5A* gene (c.2465G > A; p (Trp822Ter) in Exon 16.

Genetic testing detected the same heterozygous variant in the brother (index patient), who also had BrS, as well as in the father, who showed ECG abnormalities indicating strong penetrance of this genetic variant. The mother and sister, who had normal ECGs, were genotype negative.

## 4. Discussion

The index patient of family 1 was a carrier of a novel heterozygous splice-site variant in the *SCN5A* gene (c.4814-1G > A; p. (?) in intron 27, which has not been described in the literature so far. Since the variant is localized to the canonical splice site, it is expected that variant *SCN5A*- c.4814-1G > A causes whole exon skipping, or inclusion of an intron fragment, or exon fragment skipping. These radical changes are known to produce a loss of function of the NaV1.5 channel. Loss-of-function mutations of the NaV1.5 channel are known to cause Brugada syndrome. According to the 2015 ACMG criteria, this variant is classified as pathogenic (Class V) [8].

A subtricuspid substrate with inducible monomorphic VT originating from this region is also associated with arrhythmogenic right ventricular cardiomyopathy (ARVC), an inherited cardiomyopathy, which is characterized by cardiomyocyte loss and fibro-fatty replacement of predominantly RV myocardium [9].

Although ARVC and BrS are distinct clinical entities, a phenotypic overlap can be observed [10]. Furthermore, in some desmosomal gene-negative ARVC patients, rare *SCN5A* variants have been detected [11]. These findings demonstrate that changes in the sodium channel NaV_1.5_. might interact with other molecules or form a tighter network of interactions—also called the connexome—and serve as potential causal genetic variants in ARVC [12,13]. When assessing patients with BrS it is important to keep in mind that asymptomatic patients represent a majority of newly diagnosed patients with an incidence of arrhythmic events of 0.5% per year, and that many asymptomatic patients with BrS type I or drug-induced type I have been overtreated in the past. Yet, even if most of these patients have an uneventful course, a few may experience life-threatening ventricular arrhythmias. Therefore, their risk stratification remains challenging. A spontaneous type 1 ECG pattern, as well as other ECG markers such as an early repolarization pattern and QRS fragmentation, have been associated with a higher risk. Electrophysiological studies remain controversial in the assessment of patients with BrS. A multicenter pooled analysis showed that induction of sustained ventricular arrhythmia (VA) during electrophysiological study was associated with a higher future risk of sustained VA, and was associated with a potentially clinically meaningful outcome, particularly in asymptomatic patients with a spontaneous type 1 Brugada pattern ECG [14]. Therefore, programmed ventricular stimulation is a Class IIb (LOE B) recommendation in the latest 2022 ESC Guidelines on ventricular arrhythmias [1]. The young index patient in family 1 consulted us for a second opinion due to his spontaneous borderline Brugada type 1 ECG. He was concerned about his diagnosis and asked us to assess his individual risk for arrhythmias. So, together we decided to confirm his clinical diagnosis by flecainid challenge, genetic testing for *SCN5A* variants (Class 1 C recommendation), and further refine his arrhythmic risk by an invasive electrophysiological study. After reproducible and easy induction of rapid monomorphic VT with RV origin, we decided to map the VT origin and assess a potential substrate. We were surprised to see this small peritricuspid substrate, which is often seen in ARVC.

Family 2 and their genetic *SCN5A* (c.2465G > A; p (Trp822Ter) variant were already described in 2005 [7]. At that time, our patient (brother of the index patient) was asymptomatic except for a positive flecainide challenge test, and no further treatment was offered to him back then. He later presented with a spontaneous Brugada type 1 ECG and recurrent non-sustained VTs, suggesting a progress/manifestation of the disease that led to the implantation of a transvenous ICD. BrS manifests primarily during adulthood, with a mean age of sudden death of approximately 40 years. The youngest individual diagnosed with the syndrome was two days old and the oldest age 85 years [15]. This indicates the importance of risk stratification not only at first disease diagnosis, but also throughout the following years, since potentially life-threatening ventricular arrhythmias can occur in later adulthood as well.

The father of our index patient, also a carrier of the *SCN5A* (c.2465G > A; p (Trp822Ter) variant, who was asymptomatic in 2005 with an unremarkable 12-lead ECG at rest and a negative ajmaline challenge test back then, also showed signs of disease progression, namely conduction system disease in the latest ECG, highlighting the need for regular follow-up visits in patients and family members with a positive genotype for BrS. Our findings also emphasize that this variant not only presents as BrS and survived sudden cardiac arrest, but also conduction system disease.

Variants in *SCN5A* as a cause of hereditary conduction system dysfunction were first described in 1999 [16]. The conduction abnormalities observed in *SCN5A*^E558X/+^ pigs functionally phenocopied hereditary progressive cardiac conduction disorders (PCCD), and the pigs exhibited conduction slowing at an early age that progressively worsened with maturity, as evidenced by an age-dependent sensitivity to flecainide [17], similar to reported PCCD patients with loss-of-function variants [18]. This finding could explain the new occurrence of conduction system disease with increasing age in this patient, which has not been described with this *SCN5A* variant (c.2465G > A; p (Trp822Ter)) so far.

To date, nearly a quarter of BrS patients were found to be *SCN5A* variant carriers and more than 300 *SCN5A* variants were found to be associated with BrS, including missense, non-sense, nucleotide insertion/deletions, and splice-site variants. Those are usually loss-of-function variants [19]. They disturb the transmembrane ion flux balance at the end of the first phase of the action potential, which presents as ST-segment elevation on the surface ECG. When the current imbalance increases, the epicardial AP becomes significantly shorter and the phase 2 bipolar state becomes unbalanced with reentry, leading to the occurrence of VT and VF [20].

A decrease in sodium channel density on cell membranes leads also to a decreased current in phase 0, slows down the myocardial conduction speed, and eventually results in various conduction blocks. We hypothesized that the same mutation in the *SCN5A* gene can lead either to BrS or to an isolated cardiac conduction defect.

## 5. Conclusions

In conclusion, we describe a novel pathogenic heterozygous variant associated with BrS overlapping with ARVC, as well as disease progression with a previously reported *SCN5A* variant being associated with a phenotype of BrS and conduction system disorder in two unrelated families.

## Figures and Tables

**Figure 1 jcm-11-05625-f001:**
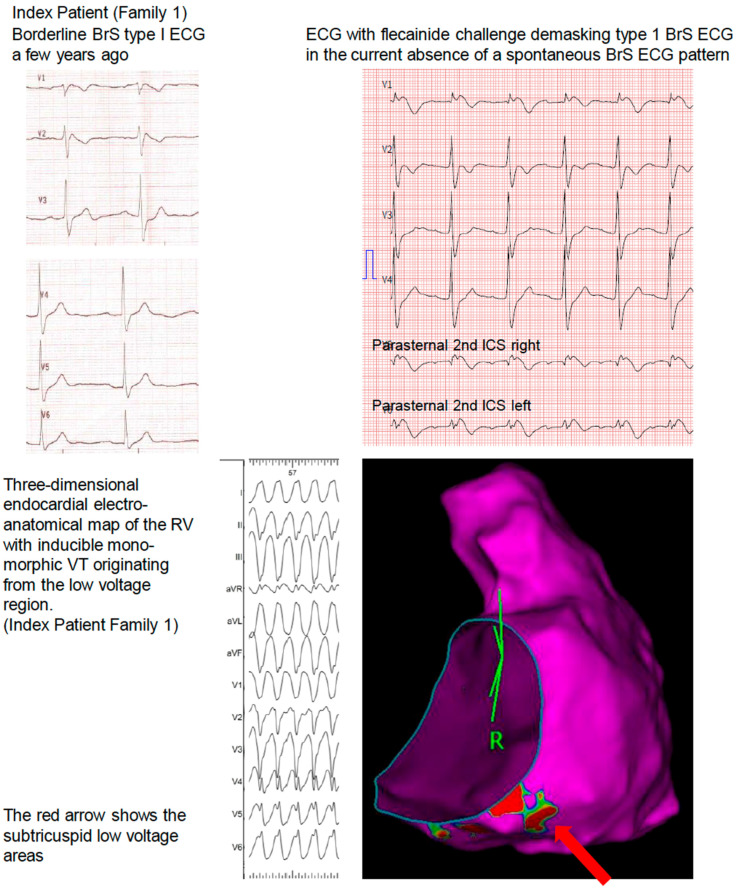
Family 1.

**Figure 2 jcm-11-05625-f002:**
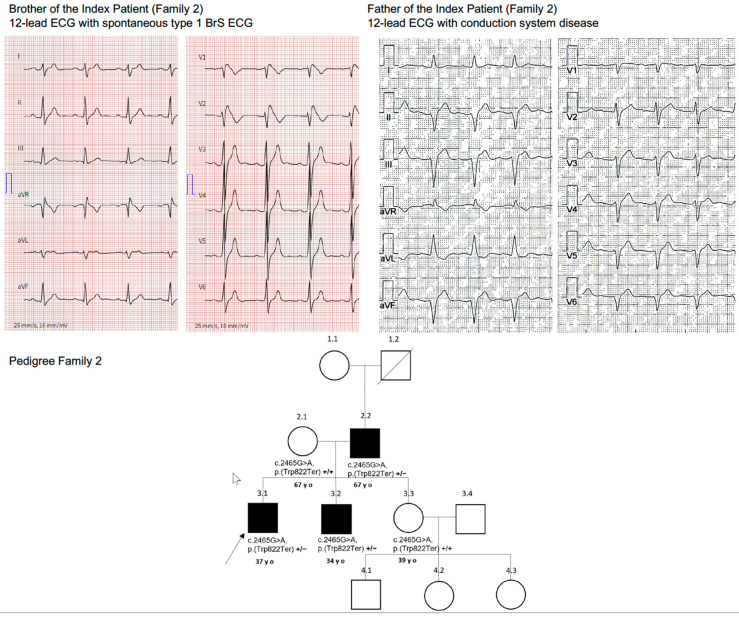
Family 2.

## Data Availability

Upon urgent request and associated need, our data are available, while our upmost intention is to protect our patients’ privacy.

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
