# Peer review of "Brugada Syndrome Associated with Different Heterozygous SCN5A Variants in Two Unrelated Families"

_jcm, 2022, doi:10.3390/jcm11195625_

Round 1

Reviewer 1 Report

Th study  reports one novel variant and one novel symptoms for an already described variant of SCN5A gene associated with Brugada Syndrome in two independent families. The manuscript is well written and discussed propoerly with inclusion of existing literature information.  The methods used for genotyping and testing are precisely described. I do not have any suggestions other than minor spellcheck throughout the manuscript.

Author Response

Reviewer 1

The study reports one novel variant and one novel symptoms for an already described variant of SCN5A gene associated with Brugada Syndrome in two independent families. The manuscript is well written and discussed propoerly with inclusion of existing literature information.  The methods used for genotyping and testing are precisely described. I do not have any suggestions other than minor spellcheck throughout the manuscript.

We thank the reviewer for his/her positive feedback. We have checked the spelling in the manuscript.

We sincerely hope that we have adequately addressed all the points and the revised version of our manuscript is now suitable for publication in Journal of Clinical Medicine

The Authors

Reviewer 2 Report

The manuscript entitled "Brugada Syndrome associated with two heterozygous SCN5A variants in two unrelated families" presented by Molito N et al. presents one novel SCN5A variant in a splice-site in intron 27 found in one man without a family history of Brugada Syndrome and another variant at the residue 822 changing a Trp to a codon stop, thus a truncated channel, at the level of domain II. While the second variant has already been reported and the pedigree of the family suggests a cause-effect relationship with the BrS, the first variant is poorly presented and the conclusion that the authors drown is precipitous. The methods section needs to be improved and also the presentation of the data.

General point

About family 1. The genetic test was performed only in the proband and with a panel of 15 genes. It is known that the number of genes suggested to be involved in BrS is much higher. Thus, the conclusion that the splice variant found in SCN5A is responsible for the BrS phenotype is precipitous.

Specific comments

Line 28: the Authors define the BrS as channelopathy. The recent literature (at least from 2020), based on the updated evidences, define the BrS a “cardiac disorder” with genetic and molecular basis. The Authors should update the definition of the syndrome looking and citing the appropriate literature.

Line 33: “unmasked by drugs”. Literature should be cited.

Lines 37-38: Please rewrite the sentence as in the present form is not clear if the two variants are present in both the families (This comment applies also to the title of the manuscript)

The Material and Method section needs improvements. In detail

Line 45: it is not specified from what kind of sample the DNA was extracted

Lines 51-53 are confusing. Please reformulate.

Paragraph 2.2.at line 69 should be part of a more detailed paragraph indicating how the subjects were clinically evaluated. I suggest to include a “Clinical Evaluation” paragraph in this section.

Also, a “Genetic analysis” paragraph would be appreciated.

Results section

Lines 89-91: this part should be included in the Material and Method section in the paragraph “Genetic analysis”

Line 91 “…a novel pathogenic…”. Please define on what basis the pathogenicity was assessed.

Lines 112-114: same as lines 89-91

Lane 105-106 vs line 118: the father is described first (lines 105-106) as asymptomatic and later (line 118) with ECG abnormalities. Even though the situation is better presented in the discussion, at this stage the presentation is confusing for the reader. Please, reformulate.

Figure 2. Pedigree family 2: As the mother is negative at the genetic screening, there is no point to show in the pedigree subjects 1.1,1.2,2.1,2.2,2.3,2.4,2.5.

Discussion

Lines 131-132: Describe better the ratio that brings towards a loss of function of the NaV1.5 channel. What is the consequence of the presented variant in intron 27 on the protein aminoacidic sequence?

Author Response

Reviewer 2

The manuscript entitled "Brugada Syndrome associated with two heterozygous SCN5A variants in two unrelated families" presented by Molito N et al. presents one novel SCN5A variant in a splice-site in intron 27 found in one man without a family history of Brugada Syndrome and another variant at the residue 822 changing a Trp to a codon stop, thus a truncated channel, at the level of domain II. While the second variant has already been reported and the pedigree of the family suggests a cause-effect relationship with the BrS, the first variant is poorly presented and the conclusion that the authors drown is precipitous. The methods section needs to be improved and also the presentation of the data.

General point

About family 1. The genetic test was performed only in the proband and with a panel of 15 genes. It is known that the number of genes suggested to be involved in BrS is much higher. Thus, the conclusion that the splice variant found in SCN5A is responsible for the BrS phenotype is precipitous.

We thank the reviewer for bringing up this important and controversial issue in changing times for genetic testing. According to the data released by ClinGen in 2019 on Brugada Syndrome gene-curation and the latest 2022-ESC guidelines in genetic testing for inherited heart diseases, the only meaniful gene in Brugada Syndrome is SCN5A. Rare variants in other genes should not be rutinely reported in diagnostic setting (Class III).  Therefore, in family 1, since the variant SCN5A- c.4814-1G>A properly explains the phenotype, there is no need to search in other “candidate-research genes” and even if we search in other genes and find something, is currently questionable to report those variants if a strong candidate SCN5A variant has been found. Nevertheless, it is important to mention, that we reported the analysis of 15 genes previously associated to Brugada Syndrome selected from a panel of 173 inherited heart diseases genes screened on the patient (list attached) no other suspected variant able to produce the phenotype was detected in this case within those 173 genes. We have cited this literature in this context.

ABCA1, ABCC9, ABCG5, ABCG8, ACADVL, ACTA2, ACTC1, ACTN2, ACVRL1, ADAMTS2, AGK, AGL, ALPK3, ANK2, ANKRD1, APOA1, APOA5, APOB, APOE, AQP1, ATP13A3, BAG3, BMPR2, CACNA1C, CACNA1D, CACNB2, CALM1, CALM2, CALM3, CASQ2, CAV1, CAV3, CDH2, COL3A1, COL5A1, COL5A2, CPT2, CRYAB, CSRP3, CTNNA3, DCHS1, DES, DMD, DPP6, DSC2, DSG2, DSP, EFEMP2, EIF2AK4, ELN, EMD, ENG, EYA4, FBN1, FBN2, FGA, FHL1, FLNA, FLNC, FOXC1, FOXE3, FOXH1, FTH1, GAA, GATA4, GATA5, GATA6, GATD1, GDF1, GDF2, GJA1, GJA5, GLA, GPD1L, GSN, HAMP, HCN4, HFE, HJV, ILK, JAG1, JPH2, JUP, KCNA5, KCND3, KCNE1, KCNE2, KCNH2, KCNJ2, KCNJ8, KCNK3, KCNQ1, LAMP2, LDB3, LDLR, LDLRAP1, LIPA, LMNA, LOX, LPL, LYZ, MYBPC3, MYH11, MYH6, MYH7, MYL2, MYL3, MYLK, MYPN, NEXN, NKX2-5, NOTCH1, NR2F2, OBSCN, PCSK9, PKP2, PLEC, PLN, PRDM16, PRKAG2, PRKD1, PRKG1, PTPN11, RANGRF, RBM20, ROBO4, RYR2, SCN10A, SCN1B, SCN2B, SCN3B, SCN4B, SCN5A, SGCD, SKI, SLC22A5, SLC2A10, SLC40A1, SLC4A3, SMAD2, SMAD3, SMAD4, SMAD9, SNTA1, SOX17, SYNE1, SYNE2, TAB2, TBX1, TBX20,TBX4, TBX5, TCAP, TECRL, TFAP2B, TFR2, TGFB2, TGFB3, TGFBR1, TGFBR2, TJP1, TLL1, TMEM43, TNNC1, TNNI3, TNNT2, TP63, TPM1, TRDN, TRPM4, TTN, TTR, VCL

Specific comments

Line 28: the Authors define the BrS as channelopathy. The recent literature (at least from 2020), based on the updated evidences, define the BrS a “cardiac disorder” with genetic and molecular basis. The Authors should update the definition of the syndrome looking and citing the appropriate literature.

We thank the reviewer for this valuable comment. We updated the definition of BrS and cite the appropriate literature as suggested.

Line 33: “unmasked by drugs”. Literature should be cited.

We thank the reviewer for this comment. We cite the appropriate literature.

Lines 37-38: Please rewrite the sentence as in the present form is not clear if the two variants are present in both the families (This comment applies also to the title of the manuscript)

We thank the reviewer for this comment. We adopted the sentence, as well as the title of the manuscript.

The Material and Method section needs improvements. In detail

Line 45: it is not specified from what kind of sample the DNA was extracted

We thank the reviewer for this comment, we have now specified that it was an EDTA blood sample.

Lines 51-53 are confusing. Please reformulate.

We thank the reviewer for this comment, we have now reformulated this part.

Paragraph 2.2.at line 69 should be part of a more detailed paragraph indicating how the subjects were clinically evaluated. I suggest to include a “Clinical Evaluation” paragraph in this section.

We thank the reviewer for this valuable comment, we have now included a “clinical evaluation” paragraph as suggested.

Also, a “Genetic analysis” paragraph would be appreciated.

We thank the reviewer for this comment. We have adapted our genetic testing paragraph and expanded it in a genetic analysis paragraph.

Results section

Lines 89-91: this part should be included in the Material and Method section in the paragraph “Genetic analysis”

We thank the Reviewer for highlighting this issue. Now we have included this part in the Material and Method section.

Line 91 “…a novel pathogenic…”. Please define on what basis the pathogenicity was assessed.

We thank the reviewer for this comment. According to the 2015 ACMG Criteria, this variant is classified as pathogenic (Class V, PVS1, PM2, PP3). We have also added the explanation in this line.

Lines 112-114: same as lines 89-91

We thank the Reviewer for highlighting this issue. We have included this part in the Material and Method section.

Lane 105-106 vs line 118: the father is described first (lines 105-106) as asymptomatic and later (line 118) with ECG abnormalities. Even though the situation is better presented in the discussion, at this stage the presentation is confusing for the reader. Please, reformulate.

We thank the reviewer for this comment. We have reformulated this part.

Figure 2. Pedigree family 2: As the mother is negative at the genetic screening, there is no point to show in the pedigree subjects 1.1,1.2,2.1,2.2,2.3,2.4,2.5.

We thank the reviewer for this comment. The reviewer is right that this pedigree subjects are irrelevant for this family, and we have deleted them.

Discussion

Lines 131-132: Describe better the ratio that brings towards a loss of function of the NaV1.5 channel. What is the consequence of the presented variant in intron 27 on the protein aminoacidic sequence?

We thank the reviewer for this comment. Since the mutation is localized to the canonical splice site, it is expected that variant SCN5A- c.4814-1G>A causes whole exon skipping, or inclusion of an intron fragment or exon fragment skipping. These radical changes are known to produce a loss of function of the NaV 1.5 channel. Loss of function mutations of the NaV1.5 channel are known to cause Brugada Syndrome. We have expanded our explanation in the revised manuscript.

We sincerely hope that we have adequately addressed all the points and the revised version of our manuscript is now suitable for publication in Journal of Clinical Medicine

The Authors

Round 2

Reviewer 2 Report

Dear Authors,

thank you for your revision of the manuscript.

Best Regards

Author Response

We thank the Reviewer for his/her positive answer.

We sincerely hope that we have adequately addressed all the points and the revised version of our manuscript is now suitable for publication in Journal of Clinical Medicine

The Authors